# Measuring health literacy: A systematic review and bibliometric analysis of instruments from 1993 to 2021

Mahmoud Tavousi[1], Samira Mohammadi[1], Jila Sadighi[1], Fatemeh Zarei[2], Ramin Mozafari Kermani[1], Rahele Rostami[1], Ali Montazeri[1,3]*

1 Health Metrics Research Center, ACECR, Iranian Institute for Health Sciences Research, Tehran, Iran,
2 Faculty of Medical Sciences, Department of Health Education, Tarbiat Modares University, Tehran, Iran,
3 Faculty of Humanity Sciences, University of Science and Culture, Tehran, Iran

* montazeri@acecr.ac.ir

**Data Availability Statement:** The authors confirm that all data underlying the findings are fully available without restriction. All relevant data are within the article.

## Abstract

### Background

It has been about 30 years since the first health literacy instrument was developed. This study aimed to review all existing instruments to summarize the current knowledge on the development of existing measurement instruments and their possible translation and validation in other languages different from the original languages.

### Methods

The review was conducted using PubMed, Web of Science, Scopus, and Google Scholar on all published papers on health literacy instrument development and psychometric properties in English biomedical journals from 1993 to the end of 2021.

### Results

The findings were summarized and synthesized on several headings, including general instruments, condition specific health literacy instruments (disease & content), population-specific instruments, and electronic health. Overall, 4848 citations were retrieved. After removing duplicates (n = 2336) and non-related papers (n = 2175), 361 studies (162 papers introducing an instrument and 199 papers reporting translation and psychometric properties of an original instrument) were selected for the final review. The original instruments included 39 general health literacy instruments, 90 condition specific (disease or content) health literacy instruments, 22 population- specific instruments, and 11 electronic health literacy instruments. Almost all papers reported reliability and validity, and the findings indicated that most existing health literacy instruments benefit from some relatively good psychometric properties.

### Conclusion

This review highlighted that there were more than enough instruments for measuring health literacy. In addition, we found that a number of instruments did not report psychometric

**Funding:** The author(s) received no specific funding for this work.

**Competing interests:** The authors have declared that no competing interests exist.

properties sufficiently. However, evidence suggest that well developed instruments and those reported adequate measures of validation could be helpful if appropriately selected based on objectives of a given study. Perhaps an authorized institution such as World Health Organization should take responsibility and provide a clear guideline for measuring health literacy as appropriate.

## Introduction

The term 'health literacy' was first used in 1974 in a paper entitled 'health education as a social policy' [1]. Since then, health literacy appeared more frequently in the biomedical literature and believed that it goes beyond the ability to read, write, and understand the meanings of words and numbers in health care settings [2]. The World Health Organization (WHO) defined health literacy as: 'cognitive and social skills that determine the motivation and ability of individuals to access understand and use the information to promote and maintain optimal health' [3]. Later the WHO regional office for Europe defined health literacy as: 'Health literacy is linked to literacy and entails people's knowledge, motivation and competences to access, understand, appraise and apply health information in order to make judgments and take decisions in every- day life concerning health care, disease prevention and health promotion to maintain or improve quality of life during the life course' [4].

Health literacy is believed to have a vital impact on public health through access to and use of health services [5, 6]. Low health literacy is associated with poor health status [6, 7], frequent use of health services, longer hospital length of stay [5, 6], and high mortality [7, 8]. In addition, some studies have linked low health literacy to unhealthy behaviors, such as smoking [4, 9–12], low physical activity [10–12], and low use of preventive services [4, 7, 10]. Essentially, health literacy plays a role in improving health outcomes both at the individual level (reducing health inequalities) and at the societal level (continuous development of health policies) [13].

Therefore, measuring health literacy is fundamental and needs appropriate measures. Among health literacy instruments, the Rapid Assessment of Adult Literacy in Medicine (REALM) [14], the Test of Functional Health Literacy (TOFHLA) [15], and the Newest Vital Sign (NVS) [16] have a long history of application. These instruments have been criticized for a number of reasons, including evaluation of only a few areas of health literacy, inadequacy for use in interventional studies, or lack of development with a health promotion perspective. In addition, most of these scales were developed and used in clinical settings [17].

In a review of the literature from 1999 to 2013, 51 instruments were identified. Of these, 26 were general health literacy instruments, 15 were condition specific (disease or content), and 10 were health literacy instruments in a specific population [18]. In a review by O'Neil et al. on self-administered health literacy instruments, 35 measures were reported (27 original; 8 derivative instruments) [19]. Nguyen et al., in their study, stated that there are more than 100 health literacy instruments, but only a small number of them have been developed using modern guidelines [20]. In addition, there were further review papers with limited focus covering either general measures or papers that reviewed condition and population- specific health literacy measures. A chronological list of selected review papers is provided in Table 1 [20–38]. However, none of the previous reviews assess instruments comprehensively. Thus, to provide insight into the literature, we performed a bibliometric analysis from the start to the end of 2021 to comprehensively review all existing instruments. We thought this might help synthesize evidence and provide a platform for investigators with similar interests to easily select, apply, or appraise an instrument when needed.

**Table 1. Review papers on health literacy instruments.**

| Author [ref.] | Year | Number of instruments reviewed | focus |
|---|---|---|---|
| Machado et al. [21] | 2014 | 4 | Health literacy in elderly hypertensive patients |
| Dickson-Swift et al. [22] | 2014 | 32 | Oral health literacy |
| O'Connor et al. [23] | 2014 | 13 | Mental health literacy |
| Parthasarathy et al. [24] | 2014 | 13 | Oral health literacy |
| Perry [25] | 2014 | 5 | Health literacy in adolescents |
| Wei et al. [26] | 2015 | Validated measures: knowledge (14), stigma (65), help-seeking related (10) | Mental health literacy (knowledge, stigma, help-seeking related) |
| Duell et al. [27] | 2015 | 43 | Health literacy in a clinical setting |
| Stonbraker et al. [28] | 2015 | 19 | Health literacy among Spanish speakers in clinical or research settings |
| Nguyen et al. [20] | 2015 | Instruments (109): General HL (58), specific content/context (51) | Health literacy measures for ethnic minority populations |
| Wei et al. [29] | 2017 | 12 | Mental health literacy tools measuring help-seeking |
| Lee et al. [30] | 2017 | 13 | Health literacy for people with diabetes |
| Shum et al. [31] | 2018 | Asthma (40), COPD (22), Asthma/COPD (3) | Airway diseases and health literacy measurement tools |
| Guo et al. [32] | 2018 | 29 | Children and adolescents |
| Wei et al. [33] | 2018 | 101 | Mental health literacy measurement tools (the stigma of mental illness) |
| Okan et al. [34] | 2018 | 15 | Health literacy instruments used in children and adolescents |
| Estrella et al. [35] | 2020 | 17 | Health literacy among US African Americans and Hispanics/Latinos with type 2 diabetes |
| Slatyer et al. [36] | 2020 | 3 | Self-reported instruments to assess health literacy in older adults |
| Ghaffari et al. [37] | 2020 | 21 | Oral and dental health literacy |
| Mafruhah et al. [38] | 2021 | 48 | Health literacy for medication use |

## Materials and methods

### Search engine and time period

The electronic databases searched included PubMed, Scopus, Web of Science, and Google Scholar. The aim was to review all full publications in biomedical journals between 1993 and 2021. The search was updated twice: once in January 2022 and once in early February 2022. The year 1993 was chosen since the first standard instrument was reported in 1993.

### Search strategy

The search strategy was limited to health literacy instruments whose psychometric information was accurately and transparently presented. Papers were retrieved using different combinations of keywords and MeSH terms including; 'Health literacy', 'eHealth literacy', 'e-Health literacy', 'e Health literacy', 'electronic Health literacy', 'Tool', 'Instrument', 'Scale', 'Questionnaire', 'Measure' and 'Inventory' in the title and abstract of papers.

All potentially relevant publications were extracted and reviewed independently by two authors (SM and FZ). Discrepancies between authors were resolved by consensus with the first investigator (MT). Then, qualified studies were obtained for full-text screening. The three authors extracted the data in order to identify eligible studies. After the final evaluation, the required data were extracted and recorded.

### Ethics statement

The Iranian Academic Center for Education, Culture, and Research (ACECR) approved the study (Code of Ethics approved: IR.ACECR.IBCRC.REC.1397.014).

## Selection criteria

This study included all original papers reporting psychometric properties of health literacy (and e-health literacy) instruments published in English. Papers only published in journals remained in the study, and books and pamphlets, dissertations, papers presented at conferences, etc., were excluded. All publications were screened using the PRISMA guideline [39].

## Quality assessment

The quality of papers was evaluated using the Consensus-based Standards for the selection of the health status Measurement Instrument (COSMIN) checklist. The COSMIN initiative aims to improve the selection of health measurement instruments [40]. For the purpose of this review reporting, six criteria (with at least eight items) were considered sufficient, and for each reported item, a score of 1 was assigned, giving a total score of 8. The criteria were reporting: internal consistency, stability (interclass correlation), face/content validity, structural validity (exploratory and confirmatory factor analyses), criterion validity, hypotheses testing (convergent or divergent validity, discriminant or known groups comparison). Then, the quality of psychometric reporting of each measure was categorized as: poor ($< 2$), fair (2, 3), good (4, 5), and excellent ($\geq 6$).

## Data synthesis

The data for each paper were extracted and summarized. The summary then was tabulated by a topic. The following information was provided: author(s)' name, year of publication, validity, and reliability, and type of instruments, including: 'general health literacy instruments', 'condition (disease or content) specific instruments', instruments that were developed for 'specific populations' [18], and e-Health Literacy instruments.

## Results

### Descriptive findings

The study flowchart is presented in Fig 1. Overall, 4848 papers were identified. After removing duplicates (n = 2336) and irrelevant documents (n = 2175), 361 papers were included in the final review. Of these, 162 papers introduced an instrument, and 199 papers reported translation and psychometric properties for an original measure. Indeed, the original instruments are briefly described in four categories in the following sections.

### General health literacy instruments

There were 39 instruments for measuring general health literacy. Historically among the general instruments, the most frequently used instruments were the REALM [14], the TOFHLA [15], and the NVS [16]. However, recently two well-developed instruments were introduced: The Health Literacy Questionnaire (HLQ) [55] and the Health Literacy Survey Questionnaire (HLS-EU-Q) [56]. The HLS-EU-Q and its newer versions [61, 69] have been widely used in European and Asian settings. Overall proper psychometric properties were reported for measures in this category. A summary of findings is presented in Table 2.

### Condition (disease or content) specific instruments

There were 90 condition specific (disease & content) instruments. Measuring health literacy for chronic non-communicable diseases, especially diabetes mellitus, has been considered more frequently. At least nine instruments assess health literacy in diabetes. Infectious diseases

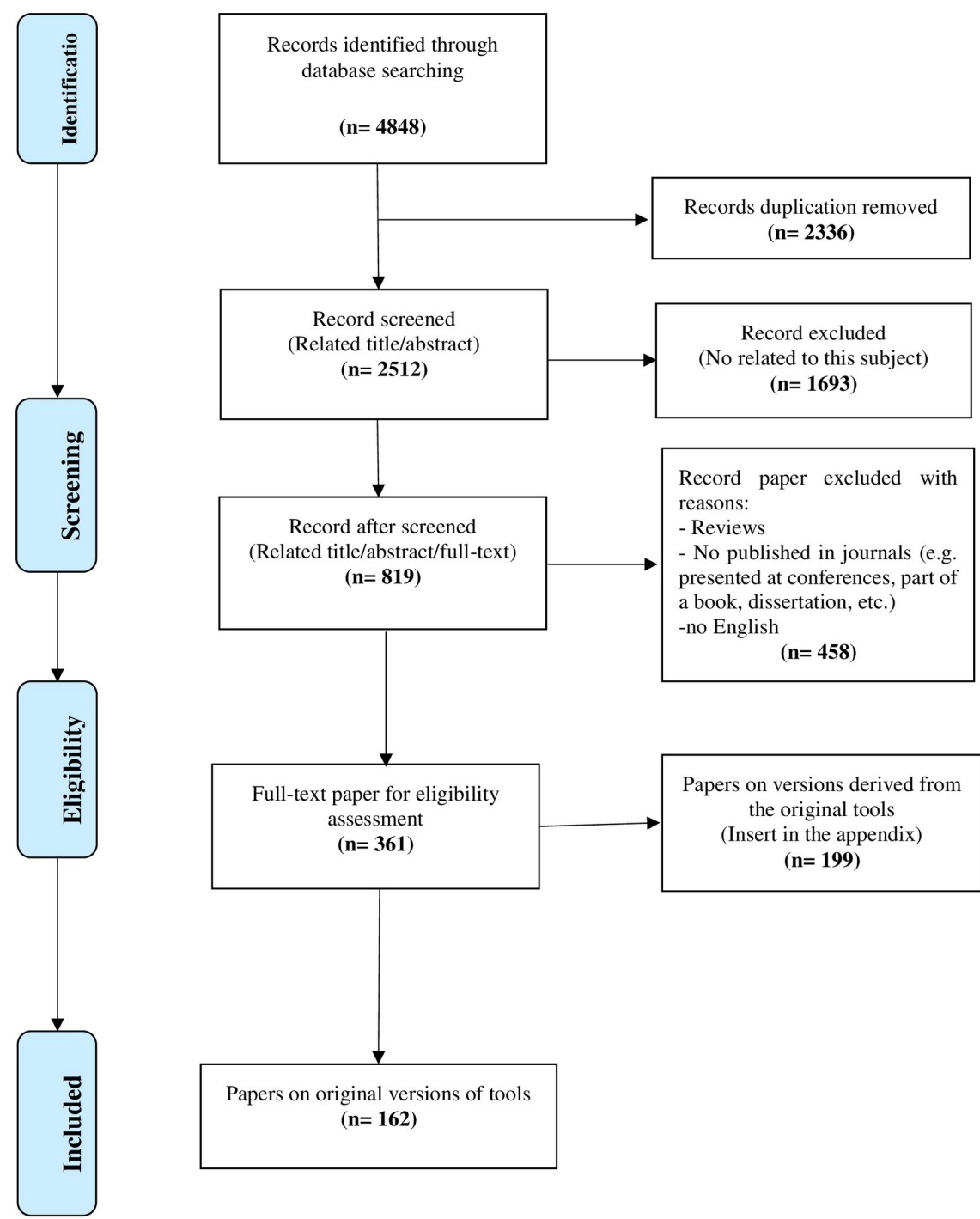

**Fig 1. Flow diagram of the study selection process.**

(such as HIV, HPV, tuberculosis, cholera, and infectious disease-specific) were the second topic of interest in developing health literacy measures. These instruments have also been well-reviewed and validated in relevant studies in terms of validity and reliability (Table 3).

Among the instruments with special content, the most frequently used were oral/dental health literacy and mental health literacy. The parental and maternal, insurance, occupational,

**Table 2. General health literacy instruments (1993–2021).**

| Author [ref.] | Year | Name (abbreviation) | Country/sample | Items | Validity | | Reliability | |
|---|---|---|---|---|---|---|---|---|
| | | | | | Face/ Content | Construct | Internal consistency | External/ Relative |
| Davis et al. [14] | 1993 | Rapid estimate of adult literacy in medicine (REALM) | American public health and primary care settings | 66 | - | Concurrent | Cronbach α = 0.86 | Test-retest = 0.99 |
| Parker et al. [15] | 1995 | Test of Functional Health Literacy in Adults (TOFHLA) | American adults patients | 57 | ✓ | Concurrent | Cronbach α = 0.98 | - |
| Baker et al. [41] | 1999 | Short form of the Test of Functional Health Literacy in Adults (S-TOFHLA) | American English speaking patients | 40 | - | Concurrent | Cronbach α = 0.97 | - |
| Weiss et al. [16] | 2005 | Newest Vital Sign (NVS) | American adults | 6 | - | Concurrent | Cronbach α = 0.78 | - |
| Lee et al. [42] | 2006 | Short Assessment of Health Literacy for Spanish-speaking Adults (SAHLSA-50) | American Spanish-speaking adults | 50 | - | Convergent; Predictive; CFA | Cronbach α = 0.92 | Test-rest = 0.86 |
| Morris et al. [43] | 2006 | Single Item Literacy Screener (SILS) | American adults with diabetes | 1 | - | Criterion | - | - |
| Zikmund-Fisher et al. [44] | 2007 | Subjective Numeracy Scale (SNS) | American general population | 8 | - | Predictive | - | - |
| Ishikawa et al. [45] | 2008 | Functional, Communicative, and Critical Health Literacy (FCCHL) | Japanese diabetic patients | 14 | - | Discriminant; EFA | Cronbach α = 0.65–0.84 | - |
| Chew et al. [46] | 2008 | 3 health literacy screening questions | American adult patients | 3 | - | Criterion | - | - |
| Pleasant et al. [47] | 2008 | Public health literacy knowledge scale | Mexican & Chinese & Ghanaian & Indian participants | 16 | ✓ | Discriminate | Cronbach α = 0.79 | - |
| Rawson et al. [48] | 2009 | Medical Term Recognition Test (METER) | American adult patients | 40 | - | Predictive | Cronbach α = 0.93 | - |
| Zhang et al. [49] | 2009 | Functional Health Literacy Tests (FHLTs) | Singapore: general public and rheumatic patients | 21 | - | Divergent (Discriminant); Convergent | Cronbach α = 0.72, 0.68 | Test-retest = 0.56; ICC = 0.95 |
| McCormack et al. [50] | 2010 | Health literacy skills instrument | American population | 25 | ✓ | CFA; Concurrent | Cronbach α = 0.86; Item-total correlation = 0.27–0.59 | - |
| Yu Ko et al. [51] | 2012 | Health Literacy Test for Singapore (HLTS) | Singapore adults | 25 | ✓ | Convergent; Predictive | Cronbach α = 0.87 | - |
| Begoray et al. [52] | 2012 | Self-reported health literacy scale | Canadian adults | 9 | - | Criterion | Cronbach α = 0.83 | - |
| Kaphingst et al. [53] | 2012 | Health literacy INDEX: health literacy demands of health information materials | American adults | 63 | - | Concurrent | - | kappa value = 0.6–0.64 |
| Helitzer et al. [54] | 2012 | The TALKDOC health literacy measurement tool | New Mexico female adults | 80 | ✓ | Convergent | - | - |
| Osborne et al. [55] | 2013 | Health Literacy Questionnaire (HLQ) | Australian general population | 44 | ✓ | CFA; Discriminant | Cronbach α = 0.86–0.90 | - |
| Sorensen et al. [56] | 2013 | Health Literacy Survey Questionnaire (HLS-EU-Q-47) | English/Bulgarian/ Dutch/German/Greek/ Polish/Spanish/Irish/ Austrian adults | 47 | ✓ | EFA | Cronbach α = 0.51–0.91 | - |
| Suka et al. [57] | 2013 | 14-item Health Literacy Scale (HLS-14) | Japanese adults | 14 | - | EFA; CFA | Cronbach α = 0.76–0.85 | - |

*(Continued)*

**Table 2.** (Continued)

| Author [ref.] | Year | Name (abbreviation) | Country/sample | Items | Validity | | Reliability | |
|---|---|---|---|---|---|---|---|---|
| | | | | | Face/ Content | Construct | Internal consistency | External/ Relative |
| Farin et al. [58] | 2013 | Health Education Literacy of Patients (HELP questionnaire) | German patient adults | 18 | ✓ | EFA; CFA; IRT | Cronbach α = 0.88–0.95 | - |
| Jordan et al. [59] | 2013 | The Health Literacy Management Scale (HeLMS) | Australian adults | 29 | ✓ | EFA; CFA | Cronbach α> 0.82 | ICC> 0.90 |
| Sand-Jecklin [60] | 2014 | Brief Health Literacy Screen (BHLS) | American adult patients | 5 | - | EFA; Concurrent | Cronbach α = 0.79 | - |
| Pelikan et al. [61]* | 2014 | Short versions of the European Health Literacy Survey Questionnaire (HLS-EU-Q16, Q6) | English/Bulgarian/ Dutch/German/Greek/ Polish/Spanish/Irish/ Austrian adults | 16 & 6 | ✓ | CFA; Concurrent | Cronbach α = 0.80 for Q6 | - |
| Kang et al. [62] | 2014 | Korean Health Literacy Instrument (KHLI) | Korean adults | 18 | ✓ | EFA; CFA | Cronbach α = 0.82 | Test-retest = 0.89 |
| Nakagami et al. [63] | 2014 | Japanese Functional Health Literacy Test (JFHLT) | Japanese adults | 16 | ✓ | Convergent; Concurrent | Cronbach α = 0.81 | - |
| Chau et al. [64] | 2015 | Chinese Health Literacy Scale for Low Salt Consumption-Hong Kong population (CHLSalt-HK) | Hong Kong older adults | 49 | ✓ | Discriminant; EFA; CFA; Concurrent t; Predictive | Cronbach α = 0.79 | Test-retest = 0.84; ICC = 0.7 |
| Haghdoost et al. [65] | 2015 | Iranian Health Literacy Questionnaire (IHLQ) | Iranian adults | 36 | ✓ | EFA | Cronbach α = 0.71–0.96 | Test-retest [ICC] = 0.73 to 0.86 |
| Zotti et al. [66] | 2017 | Single question on Self-rated Reading Ability (SrRA) | Italian adult cancer patients | 1 | ✓ | Convergent; Discriminant | - | - |
| Tsubakita [67] | 2017 | Functional Health Literacy Scale for Young Adults (funHLS-YA) | Japanese Young Adults | 19 | - | Criterion; EFA | Cronbach α = 0.75 | - |
| Kim [68] | 2017 | short version of the Korean Functional Health Literacy Test (S-KHLT) | Korean nursing students and older adults | 8 | - | Convergent; | KR-20 = .84 | - |
| Finbraten et al. [69] | 2018 | Short version of the European Health Literacy Survey Questionnaire (HLS-Q12) | Norwegian adults | 12 | - | Rasch model; CFA; Convergent | Person separation Reliability = 0.75–0.82 | - |
| Pleasant et al. [70] | 2018 | Calgary charter on health literacy scale | American general population | 5 | - | Discriminant | Cronbach α = 0.80 | - |
| Duong et al. [71] | 2019 | European Health Literacy Survey questionnaire (HLS-SF12) | Indonesian/Kazakh/ Russian/Malay/ Myanmar/Burmese/ Mandarin/Vietnamese adults | 12 | - | Convergent; CFA | Cronbach α = 0.85 | - |
| Mc Clintock et al. [72] | 2020 | Eight health literacy questions based on the national academy of medicine | Sub-Saharan Africa countries adults | 8 | ✓ | Discriminant; EFA | Cronbach α = 0.72 | - |
| Leung et al. [73] | 2020 | Rapid Estimate of Inadequate Health Literacy for older adults (REIHL) | Hong Kong patients with chronic illnesses | 12 | - | Concurrent | Sensitivity and specificity (by ROC curve analysis) | - |
| Shannon et al. [74] | 2020 | Health Communication Questionnaire (HCQ) | Australian mining industry workers | 14 | ✓ | - | - | Test-retest = 0.72 |
| Tavousi et al. [75] | 2020 | Health Literacy Instrument for Adults (HELIA) | Iranian adults | 33 | ✓ | EFA | Cronbach α = 0.72–0.89 | - |

(*Continued*)

**Table 2.** (Continued)

| Author [ref.] | Year | Name (abbreviation) | Country/sample | Items | Validity | | Reliability | |
|---|---|---|---|---|---|---|---|---|
| | | | | | Face/Content | Construct | Internal consistency | External/Relative |
| Park et al. [76] | 2021 | Korean Health Literacy Instrument | Late School-Aged Children | 16 | ✓ | EFA, CFA, Criterion | KR-20 = 0.85, 0.88, 0.82 & item-total correlations = 0.31–0.69 | - |

*Unpublished (conference).

complementary, and alternative medicine, the responsiveness of primary care practices, weight-specific childhood, overweight, social determinants of health, and non-specific neck pain health food, were other specific content measures (Table 4).

### Population- specific instruments

A number of health literacy instruments were designed for a specific population- or specific demographic population (n = 22). The grouping was based on age (adolescents, adults/elderly adults, and the elderly) or nationality (Korean, Taiwanese, English, Spanish, American, Switzerland, Australian, German, Chinese, Iranian, and Finnish). A list of instruments and their psychometric properties are shown in Table 5.

### e-Health literacy instruments

There were 11 electronic health literacy instruments. Of these, the instrument developed by Norman et al. [189] was used more frequently in various studies. A list of instruments is presented in Table 6.

### Other versions that reported for an original instrument

There were a number of instruments that translated and validated in other nations with different demographic backgrounds (n = 199). A list of these instruments is presented in Table 7.

### Results for quality assessment

As indicated in the methods section, all papers under review were assessed for quality. The results are shown in Table 8.

### Synthesis of findings

Numerous instruments have been developed during the past thirty years for measuring health literacy. This review could provide information on 162 instruments. Of these, there were two well-developed instruments:

1. HLQ, which avoided the use of prevailing theories until the later development process, and great care was taken to fully understand the experiences and lives of people, professionals, and healthcare providers [55].

2. HLS-EU-Q47, which used conceptual-based, multi-faceted attributes [56].

However, they reported limited psychometric properties. Of the remaining instruments, 15 instruments reported proper psychometric properties needed. In addition, there were a number of instruments that were translated and validated to other languages more frequently. A list of instruments is presented in Table 9.

**Table 3. Disease specific health literacy instruments (1993–2021).**

| Author [ref.] | year | Name (abbreviation) | Country/ sample | Disease | Items | Validity | | Reliability | |
|---|---|---|---|---|---|---|---|---|---|
| | | | | | | Face/ Content | Construct | Internal consistency | External |
| Huizinga et al. [77] | 2008 | Diabetes Numeracy Test (DNT43, 15) | English patients | Type 2 diabetes | 43 & 15 | ✓ | Discriminant; Convergent; EFA | KR-20 = 0.95 & 0.90 | - |
| Kim et al. [78] | 2012 | High Blood Pressure-focused Health Literacy Scale (HBP-HLS) | Korean American elder (aged 60 or older) | High blood pressure | 30 | ✓ | Convergent; Discriminant | KR-20 = 0.98 | - |
| Leung et al. [79] | 2013 | Chinese Health Literacy Scale for Diabetes (CHLSD) | Chinese patients elder (aged 65 or older) | Type 2 diabetes | 34 | ✓ | Discriminant; CFA | Cronbach α = 0.65–0.88 | Test-retest = 0.89 |
| Leung et al. [80] | 2013 | Chinese Health Literacy scale for Chronic Care (CHLCC) | Chinese patients elder (aged 65 or older) | Chronic illnesses (hypertension, diabetes mellitus, chronic obstructive pulmonary disease, or arthritis) | 24 | ✓ | Discriminant | Cronbach α = 0.91 | Test-retest (ICC) = 0.77 |
| Ownby et al. [81] | 2013 | Brief computer-administered HIV-related Health Literacy Scale (HIV-HL) | American physicians | Treated for HIV infection | 19 | - | Convergent; Concurrent; EFA | Cronbach α = 069 | - |
| Sun et al. [82] | 2013 | Skills-based instrument on health literacy regarding respiratory infectious diseases | Chinese patients | Respiratory infectious diseases | 30 | - | EFA; CFA | Cronbach α = 0.86; Item-total relation = 0.86 | - |
| Han et al. [83] | 2014 | Assessment of Health Literacy in Cancer screening (AHL-C) | Korean American immigrant women | Breast and cervical cancer screening | 52 | ✓ | Convergent; Concurrent; Discriminant | Cronbach α = 0.96; Item-total correlations = 0.18–0.86 | - |
| Dumenci et al. [84] | 2014 | Cancer Health Literacy Test (CHLT-30) & (CHLT-6) | American English speaking adults | Cancer | 30 & 6 | ✓ | CFA; Discriminant | Cronbach α = 0.88 | Test-retest = 0.90 (for CHLT-30) |
| Londono et al. [85] | 2014 | Tool for asthma patients in the Italian-speaking | Italian-speaking patient's region of Switzerland | Asthma | 19 | ✓ | - | - | ICC = 0.97 |
| Shih et al. [86] | 2016 | Health literacy questionnaire for Taiwanese hemodialysis patients | Taiwanese adult patients | Hemodialysis | 26 | ✓ | CFA | Cronbach α = 0.81 | - |
| Matsuoka et al. [87] | 2016 | Heart Failure-specific Health Literacy scale (HF-specific HL) | Japanese patients adults with HF | Heart failure | 12 | ✓ | EFA; Discriminant | Cronbach α = 0.71 | Test-retest (ICC) = 0.88–0.89 |
| Tian et al. [88] | 2016 | Infectious Disease-Specific Health Literacy (IDSHL) | Chinese population adults households | Infectious disease-specific | 22 | ✓ | EFA; Discriminant | Cronbach α = 0.75–0.81; item-total correlation (<0.30) | - |
| Mafutha et al. [89] | 2017 | Hypertension Health Literacy Assessment Tool (HHLAT) | South African adult patients | Hypertension | 11 | ✓ | Concurrent | - | - |
| Tique et al. [90] | 2017 | HIV Literacy Test (HIV-LT) | Portuguese speaking patients | HIV infection | 16 & 10 | ✓ | EFA; Convergent | KR-20 = 0.87 | - |

(*Continued*)

**Table 3.** (Continued)

| Author [ref.] | year | Name (abbreviation) | Country/ sample | Disease | Items | Validity | | Reliability | |
|---|---|---|---|---|---|---|---|---|---|
| | | | | | | Face/ Content | Construct | Internal consistency | External |
| Chou et al. [91] | 2017 | Cancer Health Literacy Scale (C-HLS) | Chinese adults patients | Newly diagnosed cancer patients | 33 | ✓ | CFA; Criterion | Spearman–Brown split-half coefficient = 0.74; KR-20 = 0.82 | - |
| Yang et al. [92] | 2018 | Infectious disease-specific health literacy (IDSHL) | General population of Tibet | Infectious disease fever, diarrhea, rash, jaundice or conjunctivitis) | 25 | - | CFA; Known-groups | Cronbach α = 0.70; split-half coefficient = 0.62 | - |
| Lee et al. [93] | 2018 | Comprehensive Diabetes Health Literacy Scale (DHLS) | Korean adults | Diabetes | 14 | ✓ | Criterion; Convergent; EFA; CFA | Cronbach α = 0.91 | Test-retest (ICC) = 0.89 |
| Khazaei et al. [94] | 2018 | Heart Health Literacy Scale (HHLS) | Iranian adults | Heart health literacy | 26 | ✓ | EFA; CFA | Cronbach α = 0.88 | Test-retest = 0.81 |
| Dehghani et al. [95] | 2018 | Multidimensional Health Literacy Questionnaire for multiple sclerosis patients (MSHLQ) | Iranian patients | Multiple sclerosis | 22 | ✓ | EFA; Known-groups | Cronbach α = 0.94 | ICC = 0.96 |
| Yeh et al. [96] | 2018 | Diabetes-specific health literacy | Mandarin/ Taiwanese-speaking patients | Type 2 diabetes | 11 | ✓ | CFA | KR-20 = 0.84 | - |
| Kanga et al. [97] | 2018 | Korean Health Literacy Scale for Diabetes Mellitus (KHLS-DM) | Korean diabetic patients | Type 2 diabetes | 58 | ✓ | Rasch analysis; EFA; Criterion; CFA | Cronbach α = 0.83 | Test-retest = 0.80 |
| Tutu et al. [98] | 2019 | Household cholera-focused health literacy scale | American households urban poor | Household cholera-focused | 13 | ✓ | EFA | Cronbach α = 0.76 | - |
| Cardoso et al. [99] | 2019 | Alfabetizacao em Saude Relacionada a Adesao Medicamentosa entre Diabeticos (ASAM-D) | Brazilian diabetic patients adults | Type 2 diabetes | 18 | ✓ | - | Cronbach α = 0.77 | Kappa coefficient = 0.31–1 |
| De Sousa et al. [100] | 2019 | Instrument of the Health Literacy regarding Diabetic Foot (HLDF) | Brazilian diabetic patients adults | Diabetic foot | 18 | ✓ | Concurrent | Cronbach α = 0.73 | ICC = 0.79; Kappa< 0.60 |
| Li et al. [101] | 2019 | Chinese Health Literacy Scale for Tuberculosis (CHLS-TB) | Chinese patients | Tuberculosis | 31 | ✓ | EFA; CFA; Discriminant | Cronbach α = 0.0.82, split-half reliability = 0.78 | Test-retest = 0.95 |
| Wu et al. [102] | 2020 | Brief tool to measure melanoma-related health literacy and attitude | Chinese adolescents | Melanoma | 13 | ✓ | CFA | Spear-Brown split-half = no reported | Kappa coefficient> 0.7 |
| Martins et al. [103] | 2020 | Oral Health Literacy among Diabetics (OHL-D) | Brazilian adults | Type 2 diabetes | 30 | ✓ | - | - | Kappa coefficient> 1 |
| Echeverri et al. [104] | 2020 | Multidimensional Cancer Literacy Questionnaire (MCLQ) | American diverse populations | Cancer | 82 | - | Content; EFA; CFA; Discriminant | Cronbach α = 0.89 | - |
| Huang et al. [105] | 2020 | Health Literacy battery for three phases of Stroke (HL-3S) | Taiwanese adults patients | Stroke survivors | 30 | - | Rasch analysis | Rasch reliability coefficients = 0.86 and 0.87 | - |

(*Continued*)

**Table 3.** (Continued)

| Author [ref.] | year | Name (abbreviation) | Country/ sample | Disease | Items | Validity | | Reliability | |
|---|---|---|---|---|---|---|---|---|---|
| | | | | | | Face/ Content | Construct | Internal consistency | External |
| Rajabi et al. [106] | 2020 | Health literacy questionnaire on the most important domains of Non Communicable Diseases (NCDs) | Iranian patient | Cardiovascular diseases, diabetes, and cancer | 27 | ✓ | EFA | Cronbach α = 0.93 | - |
| Wei et al. [107] | 2021 | health literacy specific to Chronic Kidney Disease (CKD) | Taiwanese patients | Chronic kidney disease (CKD) | 17 | ✓ | CFA | KR-20 = 0.68 | - |
| Chen et al. [108] | 2021 | Health Literacy Assessment Instrument | Chinese patients | Chronic Pain | 31 | ✓ | EFA; CFA | Cronbach α = 0.93–0.97; split-half reliability = 0.91 | Test-retest = 0.93 |
| Savci et al. [109] | 2021 | Health Literacy Scale for Protection Against COVID-19 | Turkish Adults (15–30) | COVID-19 | 20 | ✓ | EFA; CFA; Criterion | Cronbach α = 0.97; item-total correlation = 0.68–0.94 | - |
| Hiltrop et al. [110] | 2021 | COVID-19 related Health Literacy in Healthcare Professionals (HL-COV-HP) | Healthcare professionals | COVID-19 | 12 | - | EFA; CFA; Convergent | Cronbach α = 0.87 | - |

## Discussion

This bibliometric review covered the literature for about thirty years. The present review extracted and reported a wide range of health literacy instruments in several sections and perhaps could be a good reference for investigators who wish to use an instrument for measuring health literacy. In addition, the current study might help to avoid adding yet another measure to a rather long list of existing instruments.

Some general health literacy instruments have multiple versions used in different languages and populations. For instance, there were 16 versions for the REALM [14], 15 versions for the NVS [16], 6 versions for the TOFHLA [15], 13 versions for the S- TOFHLA [41], and 19 versions for the HLQ [55] (Table 7). Among the general health literacy instruments the HLS-EU-Q [56], which examines health literacy in three areas (health care, health prevention, and health promotion), has a potential to be used universally.

Despite a large number of general health literacy assessment instruments and specific topics, currently having a unique and international instrument for measuring health literacy is one of the concerns of public health professionals. This study showed that one of the most widely used instruments at the international level is the European Health Literacy Survey (HLS-EU-Q) [56]. During the development process, the English version of the HLS-EU-Q simultaneously was translated into Bulgarian, Dutch, German, Greek, Polish, Spanish, Irish, Austrian [56] and in Asia into Indonesia, Kazakhstan, Malaysia, Myanmar, Taiwan, and Vietnam [292]. Also, the Taiwanese [293–296]; Norwegian [297]; Japanese [298]; Vietnamese [299] versions of this instrument have been used in various populations, making it one of the most widely used internationally. Given this instrument's relatively wide range of applications, it may be considered a prelude for producing an international instrument for measuring health literacy.

Many instruments were developed to measure health literacy among specific diseases (chronic non-communicable diseases, especially diabetes, hypertension, and cancer). With the

**Table 4. Content specific health literacy instruments (1993–2021).**

| Author [ref.] | year | Name (abbreviation) | Country/sample | Condition | Items | Validity | | Reliability | |
|---|---|---|---|---|---|---|---|---|---|
| | | | | | | Face/ Content | Construct | Internal consistency | External |
| Cormier et al. [111]* | 2006 | Health Literacy Knowledge and Experience Survey (HL-KES) | American nursing students | Knowledge and experience | 38 | ✓ | EFA | Cronbach α = 0.79, 0.76 | - |
| Sabbahi et al. [112] | 2009 | Oral Health Literacy Instrument (OHLI) | Canadian adults | Oral health literacy | 57 | ✓ | Convergent; Discriminant; Concurrent | Cronbach α = 0.89 | ICC = 0.88 |
| Kumar et al. [113] | 2010 | Health Literacy, numeracy and the Parental Health Literacy Activities Test (PHLAT) | American caregivers of infants | Parental health literacy | 10 & 20 | ✓ | Discriminant | KR-20 = 0.76 | - |
| Macek et al. [114] | 2010 | Comprehensive oral health knowledge | American low-income adults | Oral health literacy | 4 | ✓ | Criterion | Cronbach α = 0.74 | - |
| Devi et al. [115] | 2011 | Questionnaire to assess oral health literacy among college students in Bangalore city | Indian college students | Oral health literacy | 14 | - | Convergent; Predictive | Cronbach α = 0.40 | Test-retest = 0.69 |
| Mojoyinola [116] | 2011 | Maternal Health Literacy and Pregnancy Outcome Questionnaire (MHLAPQ) | All pregnant women patients | Maternal health literacy | 33 | - | - | Cronbach α = 0.81 | - |
| Loureiro et al. [117] | 2012 | Questionario de Avaliacao da Literacia em Saude Mental (QuALiSMental) | Portuguese adolescents and young people | Mental health literacy | 46 | - | EFA | Cronbach α = 0.60–0.82 | - |
| Wong et al. [118] | 2013 | Hong Kong Oral Health Literacy Assessment Task for Pediatric dentistry (HKOHLAT-P) | Speak Chinese child/parent dyads in Hong Kong | Oral health literacy | 2 | ✓ | Convergent; Predictive t; Concurrent | Cronbach α = 0.86, 0.73 | Test-retest (ICC) = 0.63 |
| Dahlke et al. [119] | 2014 | Mini Mental Status Exam (MMSE) | American English speaking older adults | Mental health literacy | 5 | ✓ | Convergent; Criterion (Predictive) | - | - |
| Jones et al. [120] | 2014 | Health Literacy in Dentistry scale (HeLD-29) | Indigenous Australians adults | Oral health literacy | 29 | ✓ | Convergent; Predictive; Discriminant; EFA | Cronbach α = 0.91 | ICC = 0.65 |
| Naghibi Sistani et al. [121] | 2014 | Oral Health Literacy for Adults Questionnaire (OHL-AQ) | Iranian adults | Oral health literacy | 17 | ✓ | Discriminant | Cronbach α = 0.72 | Test-retest (ICC) = 0.84 |
| Paez et al. [122] | 2014 | Health Insurance Literacy Measure (HILM) | American adult | Health insurance literacy | 42 | - | EFA; CFA; Convergent | Cronbach α > 0.9 | - |
| Shreffler-Grant et al. [123] | 2014 | Montana State University (MSU) CAM health literacy scale | American older adults living in rural | Complementary and alternative medicine | 21 | ✓ | Convergent; EFA | Cronbach α = 0.75 | - |
| Villanueva Vilchis et al. [124] | 2015 | Spanish Oral Health Literacy Scale (SOHLS) | Mexican adult | Oral health literacy | 29 | ✓ | Convergent | Cronbach α = 0.74 | Test-retest (ICC) = 0.76 |

*(Continued)*

**Table 4.** (Continued)

| Author [ref.] | year | Name (abbreviation) | Country/sample | Condition | Items | Validity | | Reliability | |
|---|---|---|---|---|---|---|---|---|---|
| | | | | | | Face/ Content | Construct | Internal consistency | External |
| O'Connor et al. [125] | 2015 | Mental Health Literacy Scale (MHLS) | Australian residents | Mental health literacy | 35 | ✓ | EFA; Concurrent; Discriminant | Cronbach α = 0.87 | Test-retest = 0.79 |
| Altin et al. [126] | 2015 | Health Literacy responsiveness of Primary Care practices (HLPC) | German general population | Primary care practices | 4 | - | EFA; CFA; Concurrent | Cronbach α = 0.86 | - |
| Curtis et al. [127] | 2015 | Comprehensive Health Activities Scale (CHAS) | American participants | Comprehensive health activities | 45 | - | Predictive; Convergent; CFA | Cronbach α = 0.92 | - |
| Guttersrud et al. [128] | 2015 | Maternal Health Literacy (MaHeLi) scale | Uganda adolescents patients | Maternal health literacy | 12 | - | Rasch models | Cronbach α = 0.92; Person Separation Index (PSI) = 0.82–0.90 | - |
| Stein et al. [129] | 2015 | Adult Health Literacy Instrument for Dentistry (AHLID) | Norwegian adults older | Oral health literacy | - | ✓ | Predictive | Cronbach α (= 0.98) | Test-retest = 0.81 |
| Intarakamhang et al. [130] | 2016 | Alcohol, Baccy, Coping, Diet, and Exercise Health Literacy scale (ABCDE-HL) | Thai adults | ABCDE | 64 | ✓ | EFA; CFA | Cronbach α = 0.61–0.91 | - |
| Kapoor et al. [131] | 2016 | Determination of Functional Literacy in Dentistry (DFLD) | Indian patients | Oral health literacy | 30words/ 30 items | ✓ | Convergent; Predictive | Cronbach α = 0.84 | Test-retest = 0.69 |
| Jung et al. [132] | 2016 | Multicomponent mental health literacy measure | American local public housing authority | Mental health literacy | 26 | ✓ | Groups known; EFA; CFA; Convergent | Cronbach α = 0.76–0.84; KR-20 = 0.83 | - |
| Campos et al. [133] | 2016 | Mental Health Literacy questionnaire (MHLq) | Portuguese young people | Mental health literacy | 33 | ✓ | EFA | Cronbach α = 0.84 | Test-retest (ICC) = 0.88 |
| Squires et al. [134] | 2017 | Health literacy promotion practices assessment instrument | American health care provider | Health promotion practices | 38 | ✓ | EFA | Cronbach α = 0.95 | - |
| Bjornsen et al. [135] | 2017 | Mental Health-Promoting Knowledge (MHPK-10) | Norwegian adolescents | Mental health literacy | 10 | ✓ | Groups known; EFA; CFA | Cronbach α = 0.87 | Test-retest = 0.70 |
| Moll et al. [136] | 2017 | Mental Health Literacy tool for the Workplace (MHL-W) | Canadian healthcare workers | Mental health literacy | 16 | - | Discriminant; Convergent; EFA | Cronbach α = 0.94 | - |
| Intarakamhang et al. [137] | 2017 | HL scale for Thai childhood overweight | Thai school students | Childhood overweight | 55 | - | EFA; CFA | Cronbach α = 0.70; KR-20 = 0.76; Item-total correlation coefficient = 0.2–0.8 | - |
| Matsumoto et al. [138] | 2017 | Health Literacy of Social Determinants of Health Questionnaire (HL-SDHQ) | Japanese adults | Social determinants of health | 33 | ✓ | CFA | Cronbach α = 0.92 | - |

(Continued)

**Table 4.** (Continued)

| Author [ref.] | year | Name (abbreviation) | Country/sample | Condition | Items | Validity | | Reliability | |
|---|---|---|---|---|---|---|---|---|---|
| | | | | | | Face/ Content | Construct | Internal consistency | External |
| Tsai et al. [139] | 2018 | Weight-Specific Health Literacy Instrument (WSHLI) | Taiwanese adults | Weight-Specific | ✓ | - | Convergent; Predictive; EFA; CFA | Cronbach α = 0.80 & 0.81; split-half coefficient = 0.78 & 0.81 | - |
| Lichtveld et al. [140] | 2019 | Environmental Health Literacy (EHL) | American public health students | Environmental health literacy | 42 | ✓ | EFA; CFA | Cronbach α = 0.63–0.70 | - |
| Areerak et al. [141] | 2019 | Neck pain-specific Health Behavior in Office Workers (NHBOW) | Thai office workers | Non-specific neck pain | 6 | ✓ | EFA; CFA; Discriminative | Cronbach α = 0.64, 0.53 | Test-retest (ICC) = 0.75 |
| Zhang et al. [142] | 2019 | Chinese Parental Health Literacy Questionnaire (CPHLQ) | Chinese caregivers of children (0–3 years) | Parental health literacy | 39 | ✓ | CFA | Cronbach α = 0.89; Spilt-half (Spearman-Brown coefficient) = 0.92 | Test-retest = 0.82 |
| Irvin et al. [143] | 2019 | Water Environmental Literacy Level Scale (WELLS) | Thai adults office workers | Water environmental literacy | 6 | ✓ | Criterion; Discriminative | Cronbach α = 0.51 | - |
| Wei et al. [144] | 2019 | Mental Health Literacy tool for Educators (MHL-ED) | Canadian educators | Mental health literacy | 29 | ✓ | EFA; Groups known; | Cronbach α = 0.85 | - |
| Ayre et al. [145] | 2020 | Parenting Plus Skills Index (PPSI) | Australian parents | Parenting health literacy | 13 | ✓ | CFA; Criterion | Cronbach α = 0.70 | - |
| Intarakamhang et al. [146] | 2020 | Environmental Health Literacy (EHL) | Thai village health volunteers | Environmental health literacy | 25 | ✓ | CFA | Cronbach α = 0.91–0.93 | - |
| Suthakorn et al. [147] | 2020 | Thai Occupational Health Literacy Scale- Informal Workers (TOHLS-IF) | Thai informal workers | Occupational health literacy | 38 | ✓ | EFA; CFA | Cronbach α = 0.98 | - |
| Lin et al. [148] | 2020 | Chinese Medication Literacy Measurement (ChMLM-13 &17) | Mandarin or Taiwanese adults | Medication-related health literacy | 13 & 17 | ✓ | EFA; Convergent; Discriminant | Cronbach α = 0.83, 0.78 | - |
| Taheri et al. [149] | 2020 | Maternal Health Literacy Inventory in Pregnancy (MHELIP) | Iranian pregnant women | Maternal health literacy | 48 | ✓ | EFA | Cronbach α = 0.94 | ICC = 0.96 |
| Tabacchi et al. [150] | 2020 | Food Literacy Assessment Tool (FLAT) | Italian children | Food literacy | 16 | ✓ | Discriminant; CFA | Cronbach α = 0.73 to 0.76 | - |
| Zenas et al. [151] | 2020 | Danish Mental Health Literacy Adolescents questionnaire (MeHLA) | Danish adolescents | Mental health literacy | Not indicated- | ✓ | EFA; CFA | Cronbach α = 0.82 | - |
| Taoufik et al. [152] | 2020 | Greek Oral Health Literacy measurement instrument (GROHL-20) | Greece adult patients | Oral health literacy | 20 | ✓ | Convergent | Cronbach α = 0.80 | Test-retest (ICC) = 0.95 |

(*Continued*)

**Table 4.** (Continued)

| Author [ref.] | year | Name (abbreviation) | Country/sample | Condition | Items | Validity | | Reliability | |
|---|---|---|---|---|---|---|---|---|---|
| | | | | | | Face/ Content | Construct | Internal consistency | External |
| Chao et al. [153] | 2020 | Mental Health Literacy Scale for Healthcare Students (MHLS-HS) | Taiwanese health care students | Mental health literacy | 26 | ✓ | EFA; CFA; Convergent; Discriminant; Known groups | Cronbach α = 0.70–0.87 | - |
| Sun et al. [154] | 2021 | The Comprehensive Oral Health Literacy (COHL) | Chinese general population Community health centers in Beijing(18–86 years) | Oral health literacy | 30 | ✓ | EFA; Discriminant, Concurrent | Cronbach α = 0.72 | Test-retest = 0.972 |
| Poureslami et al. [155] | 2021 | Vancouver Airways Health Literacy Tool (VAHLT) | - | Chronic airway disease (CAD) patients | 44 | ✓ | - | - | - |
| Mahmoudian et al. [156] | 2021 | Hearing health literacy in Iranian young people | Iranian young people (12–25 years) | Hearing health literacy | 22 | ✓ | - | Cronbach α = 0.65 | - |
| Simkiss et al. [157] | 2021 | Knowledge and Attitudes to Mental Health Scales (KAMHS) | Children and adolescents (13–14 years) | Mental health literacy | 50 | ✓ | EFA; CFA | Lavaan. Omega(ω) = 0.53–76 | Test-retest = 0.40–0.64 |
| Charophasrat et al. [158] | 2021 | Oral Health Literacy Questionnaire | Thai adults | Oral Health Literacy | 21 | ✓ | Known-group; Concurrent | Cronbach α = 0.87 | - |
| Karimi et al. [159] | 2021 | Sexual health literacy related to HIV/AIDS and sexually transmitted diseases | Iranian young men (19–29 years) | Sexual health literacy | 30 | ✓ | - | Cronbach α = 0.79–0.87 | ICC = 0.79–0.87 |
| Ma et al. [160] | 2021 | Reproductive health literacy questionnaire | Chinese unmarried youth (15–24 years) | Reproductive health literacy | 58 | ✓ | CFA | Cronbach α = 0.91; split-half reliability = 0.84 | Test-retest = 0.72 |
| Suto et al. [161] | 2021 | Health literacy scale for preconception care | Japanese adults (16–49 years) | Reproductive health literacy | 17 & 25 | ✓ | EFA; Criterion | Cronbach α = 0.68–0.89 & 0.82–0.90 | - |
| Kodama et al. [162] | 2021 | Mental Health Literacy Scale for Depression Affecting the Help-Seeking Process | Health Professional Students | Mental health literacy | 10 | ✓ | EFA; CFA; Criterion | Cronbach α = 0.68–0.85 | Test-retest (ICC) = 0.78 |
| Aller et al. [163] | 2021 | Mental Health Awareness and Advocacy Assessment Tool (MHAA-AT) | college attending participants of Amazon's Mechanical Turk | Mental health literacy | 65 | ✓ | EFA; Convergent | Cronbach α = 0.62–0.95 | - |
| Robbins et al. [164] | 2021 | OSA Functional Health Literacy (SOFHL) | Dwelling black participants, at risk for OSA | Obstructive sleep apnea functional health literacy | 18 | - | - | Cronbach α = 0.71–0.81 | - |
| Rabin et al. [165] | 2021 | Mental Health Literacy Assessment-college (MHLA-c) | US college students | Mental health literacy | 54 | ✓ | Known groups | KR-20 = 0.74–0.75 | - |

(*Continued*)

**Table 4.** (Continued)

| Author [ref.] | year | Name (abbreviation) | Country/sample | Condition | Items | Validity | | Reliability | |
|---|---|---|---|---|---|---|---|---|---|
| | | | | | | Face/ Content | Construct | Internal consistency | External |
| Moein et al. [166] | 2021 | Physical activity health literacy in Iranian older adults (PAHLIO) questionnaire | Iranian older adults (60–75 years) | Physical activity health literacy | 18 | ✓ | EFA; CFA | Cronbach α = 0.85–0.94 | Test-retest (ICC) = 0.89–1 |

\* Unpublished (dissertation).

widespread prevalence of chronic non-communicable, there was a strong desire to develop such instruments. As shown in Table 3, among chronic diseases, diabetes has received more attention than other diseases. Among the instruments that consider a specific content (e.g., maternal, parental, environmental, obesity, and weight gain), oral/dental health literacy and mental health literacy have received special attention.

Development and psychometric evaluation of health literacy instruments was observed in different countries. We recognized health literacy instruments in different languages such as Korean, Taiwanese, English, Spanish, American, Australian, German, Switzerland, Finnish, Iranian, Chinese, Japanese, Brazilian, Philippines, and Vietnamese. As shown in Table 5, the countries of Southeast Asia, especially China, have a long history of activity in this field. It has also been shown that the American population and the populations of Southeast Asian countries (Chinese, Taiwanese, and Koreans) address a large number of health literacy assessment instruments.

One of the unique features of this study is the reporting of e-health literacy instruments. There were eleven instruments available for measuring e-health literacy (Table 6). The existence of many different versions of such instruments (Table 7) demonstrates a growing tendency to measure health literacy related to the increasing use of interment and social media by the general public almost everywhere.

Finally, one should note that the most important question is, do we need so many instruments for measuring health literacy? Although one could not prevent investigators from developing new instruments, it is evident that such haphazard development of instruments is not helpful. It seems that we need a core global general health literacy instrument for use around the globe. Then perhaps it is possible to add a few contents/disease-specific, population- specific, or e-health literacy items to the general instruments according to their use. The experience of the European Organization for Research and Treatment of Cancer-EORTC (the Quality of Life Study Group) might be useful to be adapted (https://qol.eortc.org/quality-of-life-group/).

## Limitations

The main criterion in extracting information was the availability of the full-text papers. In cases of no access to the original text, the required information was extracted from their abstracts. Otherwise, such studies were removed from the review. In addition, we only reviewed papers that included the word health literacy in the title. Thus there is a risk of missing papers that did not use health literacy in their titles.

## Conclusion

This review highlighted that there were more than enough instruments for measuring health literacy. In addition, we found that a number of instruments did not report psychometric

**Table 5. Population- specific health literacy instruments (1993–2021).**

| Author [ref.] | year | Name (abbreviation) | Country/sample | Items | Validity | | Reliability | |
|---|---|---|---|---|---|---|---|---|
| | | | | | Face/ Content | Construct | Internal consistency | External |
| Lee TW et al. [167] | 2009 | Korean Health Literacy Scale (KHLS) | Korean older adults | 24 | ✓ | EFA; CFA | Cronbach α = 0.89 | - |
| Pan et al. [168] | 2010 | Taiwan Health Literacy Scale (THLS) | Taiwanese elderly adults | 66 | - | Concurrent; Discriminant | Cronbach α = 0.98 | - |
| Tsai et al. [169] | 2010 | Mandarin Health Literacy Scale (MHLS) | Taiwanese adults | 50 | ✓ | EFA; CFA; Convergent; Predictive | Cronbach α = 0.95; Spearman–Brown split-half coefficient = 0.95 | - |
| Weidmer et al. [170] | 2012 | Consumer Assessment of Healthcare Providers and Systems (CAHPS) | English and Spanish adult patients | 22 | - | CFA | Cronbach α = 0.89 | - |
| Massey et al. [171] | 2013 | Multidimensional measure of adolescent health literacy | American adolescent | 24 | ✓ | EFA | Cronbach α = 0.83 | - |
| Wang et al. [172] | 2014 | Multidimensional instrument to assess competencies for health | Switzerland resident population | 74 | ✓ | EFA; CFA | Cronbach α = 0.72–0.81 | - |
| Harper et al. [173] | 2014 | Health literacy assessment for young adult college students | American undergraduate student | 51 | ✓ | CFA: IRT | - | - |
| Yuen et al. [174] | 2014 | Health Literacy of Caregivers Scale- Cancer (HLCS-C) | Australian cancer caregivers | 88 | ✓ | - | - | - |
| Manganello et al. [175] | 2015 | he Health Literacy Assessment Scale for Adolescents (HAS-A) | American Teen (12–19) | 15 | ✓ | EFA; Criterion | Cronbach α = 0.73–77 | - |
| Shen et al. [176] | 2015 | Chinese resident health literacy scale | Chinese population-based | 64 | - | CFA; Discriminant | Cronbach α = 0.95; Spearman–Brown split-half coefficient = 0.94 | - |
| Abel et al. [177] | 2015 | Short survey tool for public health and health promotion research | German-speaking young adults | 8 | - | EFA; CFA; Discriminant | Cronbach α = 0.64 | - |
| Ghanbari et al. [178] | 2016 | Health Literacy Measure for Adolescents (HELMA) | Iranian adolescents | 44 | ✓ | EFA | Cronbach α = 0.93 | Test-retest (ICC) = 0.93 |
| Paakkari et al. [179] | 2016 | Health Literacy for School- Aged Children (HLSAC) | Finnish school-aged children | 10 | ✓ | CFA | Cronbach α = 0.93 | Test-retest = 0.83 |
| Yang et al. [180] | 2017 | The Health Literacy Index for Female Marriage Immigrants (HLI-FMI) | Asian women | 12 | - | CFA; Discriminant; Concurrent | Cronbach α = 0.74 | - |
| Ernstmann et al. [181] | 2017 | Health Literacy-sensitive Communication (HL-COM) | German adult patients | 9 | - | EFA; CFA | Cronbach α = 0.91; Item-total correlation = 0.622–0.762 | - |
| Chang et al. [182] | 2017 | Instrument Of Health Literacy Competencies (IOHLC) | Chinese-speaking health professionals | 49 | - | EFA; CFA; Discriminant; Convergent; IRT | Cronbach α = 0.97 | - |
| Eliason et al. [183] | 2017 | Health literacy among Lesbian, Gay, and Bisexual (LGB) | American adults | 10 | ✓ | EFA | Cronbach α = 0.95 | Test-retest = 0.91 |
| Hashimoto et al. [184] | 2017 | health Literacy Scale among Brazilian Mothers (HLSBM) | Brazilian mothers | 10 | ✓ | EFA; CFA; Concurrent | Cronbach α = 066–0.89 | - |
| Bradley-Klug et al. [185] | 2017 | Health Literacy and Resiliency Scale: Youth version (HLRS-Y) | American youth | 37 | - | EFA; Discriminant | Cronbach α = 0.88–0.94 | - |
| Guo et al. [186] | 2018 | Chinese eight-item Health Literacy Assessment Tool (c-HLAT-8) | Chinese secondary school students | 8 | ✓ | CFA; Convergent | Cronbach α = 0.94; ICC = 0.72 | - |

*(Continued)*

**Table 5.** (Continued)

| Author [ref.] | year | Name (abbreviation) | Country/sample | Items | Validity | | Reliability | |
|---|---|---|---|---|---|---|---|---|
| | | | | | Face/ Content | Construct | Internal consistency | External |
| Azizi et al. [187] | 2019 | Health Literacy Scale for Workers (HELSW) | Iranian workers | 34 | ✓ | EFA | Cronbach α = 0.90 | Test-retest = 0.69 to 0.86; ICC = 0.72 to 0.84 |
| Domanska et al. [188] | 2020 | Measurement Of Health Literacy Among Adolescents Questionnaire (MOHLAA-Q) | German adolescents | 29 | ✓ | Convergent; Concurrent; CFA | Cronbach α = 0.79 | - |

properties sufficiently. However, evidence suggest that well developed instruments and those reported adequate measures of validation could be helpful if appropriately selected based on objectives of a given study. Perhaps an authorized institution such as World Health

**Table 6.** Electronic health literacy instruments (1993–2021).

| Author [ref.] | year | Name (abbreviation) | Country/sample | Items/ Terms/ phrases | Validity | | Reliability | |
|---|---|---|---|---|---|---|---|---|
| | | | | | Face/ Content | Construct | Internal consistency | External |
| Norman et al. [189] | 2006 | The e-Health Literacy Scale (e-HEALS) | Canadian youth | 8 | ✓ | EFA | Cronbach α = 0.88 | Test-retest = 0.40–0.68 |
| Hahn et al. [190] | 2011 | Health Literacy assessment using Talking Touchscreen (Health LiTT) | American English speaking patients | 82 | ✓ | IRT; Discriminant | Cronbach α≥ 0.9 | - |
| Ownby et al. [191] | 2013 | Fostering Literacy for Good Health Today (FLIGHT) & Vive Desarollando Amplia Salud (VIDAS) | Spanish and English speaking adults | 82 | ✓ | EFA; Concurrent; Know groups | Cronbach α = 0.56–0.83 | - |
| Seçkin et al. [192] | 2016 | Electronic Health Literacy Scale (e-HLS-19) | American residents adults | 19 | - | EFA; CFA | Cronbach α = 0.93; Item total correlations = 0.09–0.81 | - |
| Van der Vaart et al. [193] | 2017 | Digital Health Literacy Instrument (DHLI) | General Dutch population | 21 | ✓ | EFA | Cronbach α> 0.68–0.88 | ICC = 0.77 |
| Kayser et al. [194] | 2018 | English/Danish version of e-Health Literacy Questionnaire (eHLQ) | English/Danish people with chronic conditions | 35 | - | IRT; EFA; CFA | Cronbach α> 0.7 | - |
| Paige et al. [195] | 2019 | Transactional e-Health Literacy Instrument (TeHLI) | American patients | 18 | - | CFA | Cronbach α = 0.90 | - |
| Woudstra et al. [196] | 2019 | Computer-based and performance-based instrument to assess health literacy skills for informed decision making in colorectal cancer screening | Dutch adults | 22 | - | IRT; CFA; Convergent; Predictive | Cronbach α = 0.66 | - |
| Castellvi et al. [197] | 2020 | Espaijove.net Mental Health Literacy test (EMHL) | Spanish adolescents | 35 | - | Groups known; Convergent | Cronbach α = 0.610 & 0.74 | Test-retest (ICC) = 0.57 & 0.42 |
| Liu et al. [198] | 2021 | eHealth Literacy Scale (eHLS-Web 3.0) | Chinese college students | 24 | ✓ | Convergent, Concurrent; EFA; CFA | Cronbach α = 0.97 | Test-retest = 0.85 |
| Duong et al. [199] | 2021 | eHealthy Diet Literacy Questionnaire (e-HDLQ) | Taiwanese adults aged 18 years and above | 11 | ✓ | EFA; Convergent | Cronbach α = 0.64 | - |

**Table 7. The original health literacy instruments and the existing translations and validation versions (1993–2021).**

**General health literacy instruments**

| Author [ref.] | Original instrument [abbreviation] | Translations | Validation and other versions |
|---|---|---|---|
| Davis et al. [14] | Rapid Estimate of Adult Literacy in Medicine (REALM) | UK [200]; Korean American [201]; Arabic [202]; | REALM-SF [203]; REAL-G [204, 205]; REAL-VS [206]; REALM-Teen [207, 208]; REALD-30 [209–211]; REALD-20 [212]; REALD-99 [213]; OHLA [214, 215] |
| Parker et al. [15] | Test of Functional Health Literacy in Adults (TOFHLA) | Serbian [216]; Danish [217]; American [218]; Albanian [219]; | TOFHLA in dentistry (TOFHLiD) [220]; OA-TOFHLiD [221] |
| Baker et al. [41] | Short form of the Test of Functional Health Literacy in Adults (S-TOFHLA) | Korean American [201]; Arabic [202, 222, 223]; Serbian [216]; Turkish [224]; Spanish [225]; Chinese [226]; Italian [66]; American [227]; Chines [228]; Hebrew [229]; English-Spanish [230] | - |
| Weiss et al. [16] | Newest Vital Sign (NVS) | American [208, 227, 231, 232]; Brazilian Portuguese [233, 234]; Italian [66, 235]; Taiwanese [236]; Brazilian [237]; UK [238]; Dutch [239]; Turkish [240]; Arabic [223, 241]; | - |
| Lee et al. [42] | Short Assessment of Health Literacy for Spanish-speaking Adults (SAHLSA-50) | Dutch [242]; Portuguese [243–245]; Dutch [246]; Spanish & English [247] | SAHLPA-33 [248] |
| Morris et al. [43] | Single Item Literacy Screener (SILS) | Arabic [202, 222, 223]; Italian [66, 249]; American [227] | - |
| Zikmund-Fisher et al. [44] | Subjective Numeracy Scale (SNS) | English-Spanish [230]; American [250] | - |
| Ishikawa et al. [45] | Functional, Communicative, and Critical Health Literacy (FCCHL) | German [251]; Dutch [252]; French [253]; Iranian [254]; Japanese [255]; Australian [256]; American [257, 258]; Korean [259]; Swedish [260]; | FCCHL-12 [261] |
| Chew et al. [46] | Health Literacy Screening Questions | English-Spanish [230]; American [262–265]; American-English and Spanish [266]; Hungary/Italy/Lebanon/Switzerland/Turkey [267] | - |
| Pleasant et al. [47] | Public Health Literacy Knowledge Scale | Turkish [268] | - |
| Rawson et al. [48] | Medical Term Recognition Test (METER( | Italian [269]; Portuguese [270] | - |
| McCormack et al. [50] | Health Literacy Skills Instrument | - | HLSI-SF-10 [271] |
| Osborne et al. [55] | Health Literacy Questionnaire (HLQ) | Danish [272]; Slovak [273]; Norwegian [274]; Ghanaian [275]; German [276]; Australian [277–280]; Chinese [281, 282]; Urdo [283]; Norwegian [284]; Yoruba [285]; Brazilian [286]; Brazilian Portuguese [287]; French [288, 289]; American [290] | - |
| Sorensen et al. [56] | European Health Literacy Survey Questionnaire (HLS-EU-Q-47) | Albanian [219]; Turkish [291]; Indonesian/Kazakh/Malay/Myanmar/Burmese/Mandarin/Vietnamese [292]; Taiwanese [293–296]; Norwegian [297]; Japanese [298]; Vietnamese [299] | - |
| Suka et al. [57] | 14-item Health Literacy Scale (HLS-14) | Brazilian Portuguese [300] | - |
| Pelikan et al. [61] | Short versions of the European Health Literacy Survey Questionnaire (HLS-EU-Q16, Q6) | Turkish [301]; Italian [302]; Icelandic [303]; French [304]; Arabic/French [305]; Swedish-Arabic [306]; Japanese [307]; Brazilian Portuguese [308]; Pakistanian [309]; German [310]; French [311] | - |
| Haghdoost et al. [65] | Iranian Health Literacy Questionnaire (IHLQ) | Iranian [312] | - |
| Finbraten et al. [69] | Short version of Health Literacy Survey Questionnaire (HLS-Q12) | Japanese [307] | - |
| Duong et al. [71] | European Health Literacy Survey Questionnaire (HLS-SF12) | Taiwanese [313]; Vietnam [314]; Turkish [315]; Japanese [307] | - |
| **Disease specific health literacy instruments** | | | |

*(Continued)*

**Table 7.** (Continued)

| Huizinga et al. [77] | Diabetes Numeracy Test (DNT-43, 15) | - | DNT-5 [230] |
|---|---|---|---|
| Kim et al. [78] | High Blood Pressure-focused Health Literacy Scale (HBP-HLS) | Chinese [316] | - |
| Leung et al. [79] | Chinese Health Literacy Scale for Diabetes (CHLSD) | Chinese [317] | - |
| Dumenci et al. [84] | Cancer Health Literacy Along a Continuum (CHLT-30) & (CHLT-6) | American [318]; Chinese [319] | - |
| Matsuoka et al. [87] | Heart Failure-specific Health Literacy scale (HF-specific HL) | Chinese [320]; Iranian [321] | - |
| **Content specific health literacy instruments** | | | |
| Cormier et al. [111] | Health Literacy Knowledge and Experience Survey (HL-KES) | Iranian [322] | – |
| Sabbahi [112] | Oral Health Literacy Instrument (OHLI) | Russian [323]; Chilean [324]; Malaysian [325] | - |
| Kumar et al. [113] | Health Literacy, Numeracy and The Parental Health Literacy Activities Test (PHLAT) | Spanish [326] | - |
| Wong et al. [118] | Hong Kong Oral Health Literacy Assessment Task for Pediatric Dentistry (HKOHLAT-P) | Brazilian-Portuguese [327] | - |
| Jones et al. [120] | Health Literacy in Dentistry scale (HeLD-29) | Thai [328]; Australian [329]; Brazilian [330, 331] | He LD-14 [332] |
| Naghibi Sistani et al. [121] | Oral health literacy for Adults Questionnaire (OHL-AQ) | American [333, 334]; Persian [335]; Hindi [336]; Mandarin [337] | - |
| Shreffler-Grant et al. [123] | Montana State University (MSU) CAM Health Literacy Scale | American [338] | - |
| O'Connor et al. [125] | Mental Health Literacy Scale (MHLS) | Pakistani [339]; South African and Zambian [340]; Arabic [341]; Chinese [342]; Portuguese [343]; Iranian [344–348]; | - |
| Jung [132] | Multicomponent Mental Health Literacy Measure (MMHLM) | - | MMHLM for Student Athletes and Therapists [349] |
| Campos et al. [133] | Mental Health Literacy (MHLq) | Portuguese [350] | - |
| Matsumoto et al. [138] | Social Determinants of Health Questionnaire (HL-SDHQ) | Korean [351] | - |
| **Population- specific health literacy instruments** | | | |
| Lee TW et al. [167] | Korean Health Literacy Scale (KHLS) | Korean [352] | - |
| Pan et al. [168] | Taiwan Health Literacy Scale (THLS) | - | STHLS [353]; THLS for Middle-Aged and Older People [354] |
| Tsai et al. [169] | Mandarin Health Literacy Scale (MHLS) | - | S-MHLS [355] |
| Yuen et al. [174] | Health Literacy of Caregivers Scale-Cancer (HLCS-C) | Australian [356] | - |
| Manganello et al. [175] | Health Literacy Assessment Scale for Adolescents (HAS-A) | Arabic [357] | - |
| Paakkari et al. [179] | Health Literacy for School-Aged Children (HLSAC) | Turkish [358]; Polish [359]; Danish [360]; Finnish/Polish/Slovak/Belgian [361] | - |
| **Electronic health literacy instruments** | | | |

(*Continued*)

**Table 7.** (Continued)

| Norman et al. [189] | e-Health Literacy Scale (e-HEALS) | Swedish-Arabic [306]; Italian [362–364]; Portuguese [365]; Dutch [366]; Hungarian [367]; Greek and Cypriot [368]; African-American and Caucasian [369]; US, UK, New Zealand [370]; UK [371]; American-Hispanic [372]; American [373–375]; Taiwanese [199]; Indonesian [376]; Polish [377]; Australian [378]; Korean [379, 380]; Arabic [381]; Iranian [382, 383]; Serbian [384]; Norwegian [385]; Ethiopian [386]; Swiss-German [387]; Brazilian [388, 389]; Chinese [390–392] | - |
|---|---|---|---|
| Hahn et al. [190] | Health Literacy Assessment Using Talking Touchscreen Technology (Health LiTT) | - | 10-item Health LiTT [393] |
| Van der Vaart et al. [193] | Digital Health Literacy Instrument (DHLI) | American [394] | - |
| Kayser et al. [194] | English/Danish version of e-Health Literacy Questionnaire (eHLQ) | Australian [395] | - |

**Table 8. The results for quality assessment of existing health literacy instruments (1993–2021).**

| Author [ref.] | Reliability | | Content & face | Validity | | | | | | Ratings |
|---|---|---|---|---|---|---|---|---|---|---|
| | | | | | | Construct | | | | |
| | | | | Structural | | Criterion | Hypothesis testing | | | |
| | Internal Consistency | Test- retest (ICC) | | EFA | CFA | Predictive & Concurrent | Convergent | Discrimination & Known groups comparison | | |
| **General health literacy instruments** | | | | | | | | | | |
| Davis et al. [14] | ✓ | ✓ | - | - | - | ✓ | - | - | | Fair |
| Parker et al. [15] | ✓ | - | ✓ | - | - | ✓ | - | - | | Fair |
| Baker et al. [41] | ✓ | - | - | - | - | ✓ | - | - | | Fair |
| Weiss et al. [16] | ✓ | - | - | - | - | ✓ | - | - | | Fair |
| Lee et al. [42] | ✓ | ✓ | - | - | ✓ | ✓ | ✓ | - | | Good |
| Morris et al. [43] | - | - | - | - | - | ✓ | - | - | | Poor |
| Zikmund-Fisher et al. [44] | - | - | - | - | - | ✓ | - | - | | Poor |
| Ishikawa et al. [45] | ✓ | - | - | ✓ | - | - | - | ✓ | | Fair |
| Chew et al. [46] | - | - | - | - | - | ✓ | - | - | | Poor |
| Pleasant et al. [47] | ✓ | - | ✓ | - | - | - | - | ✓ | | Fair |
| Rawson et al. [48] | ✓ | - | - | - | - | ✓ | - | - | | Fair |
| Zhang et al. [49] | ✓ | ✓ | - | - | - | - | ✓ | ✓ | | Good |
| McCormack et al. [50] | ✓ | - | ✓ | - | ✓ | ✓ | - | - | | Good |
| Yu Ko et al. [51] | ✓ | - | ✓ | - | - | ✓ | ✓ | - | | Good |
| Begoray et al. [52] | ✓ | - | - | - | - | ✓ | - | - | | Fair |
| Kaphingst et al. [53] | - | ✓ | - | - | - | ✓ | - | - | | Fair |
| Helitzer et al. [54] | - | - | ✓ | - | - | - | ✓ | - | | Fair |
| Osborne et al. [55] | ✓ | - | ✓ | - | ✓ | - | - | ✓ | | Good |
| Sorensen et al. [56] | ✓ | - | ✓ | ✓ | - | - | - | - | | Fair |
| Suka et al. [57] | ✓ | - | - | ✓ | ✓ | - | - | - | | Fair |
| Farin et al. [58] | ✓ | - | ✓ | ✓ | ✓ | - | - | - | | Good |
| Jordan et al. [59] | ✓ | ✓ | ✓ | ✓ | ✓ | - | - | - | | Good |
| Sand-Jecklin [60] | ✓ | - | - | ✓ | - | ✓ | - | - | | Fair |
| Pelikan et al. [61]* | ✓ | - | ✓ | - | ✓ | ✓ | - | - | | Good |

(Continued)

**Table 8.** (Continued)

| Author [ref.] | Reliability | | Content & face | Validity | | | | | | Ratings |
| | | | | Construct | | | | | | |
| | | | | Structural | | Criterion | Hypothesis testing | | | |
| | Internal Consistency | Test- retest (ICC) | | EFA | CFA | Predictive & Concurrent | Convergent | Discrimination & Known groups comparison | | |
| Kang et al. [62] | ✓ | ✓ | ✓ | ✓ | ✓ | - | - | - | | Good |
| Nakagami et al. [63] | ✓ | - | ✓ | - | - | ✓ | ✓ | - | | Good |
| Chau et al. [64] | ✓ | ✓ | ✓ | ✓ | ✓ | ✓ | - | ✓ | | Excellent |
| Haghdoost et al. [65] | ✓ | ✓ | ✓ | ✓ | - | - | - | - | | Good |
| Zotti et al. [66] | - | - | ✓ | - | - | - | ✓ | ✓ | | Fair |
| Tsubakita et al. [67] | ✓ | - | - | ✓ | - | ✓ | - | - | | Fair |
| Kim [68] | ✓ | - | - | - | - | - | ✓ | - | | Fair |
| Finbraten et al. [69] | ✓ | - | - | - | ✓ | - | ✓ | - | | Fair |
| Pleasant et al. [70] | ✓ | - | - | - | - | - | - | ✓ | | Fair |
| Duong et al. [71] | ✓ | - | - | - | ✓ | - | ✓ | - | | Fair |
| Mc Clintock et al. [72] | ✓ | - | ✓ | ✓ | - | - | - | ✓ | | Good |
| Leung et al. [73] | - | - | - | - | - | ✓ | - | - | | Poor |
| Shannon et al. [74] | - | ✓ | ✓ | - | - | - | - | - | | Fair |
| Tavousi et al. [75] | ✓ | - | ✓ | ✓ | - | - | - | - | | Fair |
| Park et al. [76] | ✓ | - | ✓ | ✓ | ✓ | ✓ | - | - | | Good |
| **Disease specific health literacy instruments** | | | | | | | | | | |
| Huizinga et al. [77] | ✓ | - | ✓ | ✓ | - | - | ✓ | ✓ | | Good |
| Kim et al. [78] | ✓ | - | ✓ | - | - | - | ✓ | ✓ | | Good |
| Leung et al. [79] | ✓ | ✓ | ✓ | - | ✓ | - | - | ✓ | | Good |
| Leung et al. [80] | ✓ | ✓ | ✓ | - | - | - | - | ✓ | | Good |
| Ownby et al. [81] | ✓ | - | - | ✓ | - | ✓ | ✓ | - | | Good |
| Sun et al. [82] | ✓ | - | - | ✓ | ✓ | - | - | - | | Fair |
| Han et al. [83] | ✓ | - | ✓ | - | - | ✓ | ✓ | ✓ | | Good |
| Dumenci et al. [84] | ✓ | ✓ | ✓ | - | ✓ | - | - | ✓ | | Good |
| Londono et al. [85] | - | ✓ | ✓ | - | - | - | - | - | | Fair |
| Shih et al. [86] | ✓ | - | ✓ | - | ✓ | - | - | - | | Fair |
| Matsuoka et al. [87] | ✓ | ✓ | ✓ | ✓ | - | - | - | ✓ | | Good |
| Tian et al. [88] | ✓ | - | ✓ | ✓ | - | - | - | ✓ | | Good |
| Mafutha et al. [89] | - | - | ✓ | - | - | ✓ | - | - | | Fair |
| Tique et al. [90] | ✓ | - | ✓ | ✓ | - | - | ✓ | - | | Good |
| Chou et al. [91] | ✓ | - | ✓ | - | ✓ | ✓ | - | - | | Good |
| Yang et al. [92] | ✓ | - | - | - | ✓ | - | - | ✓ | | Fair |
| Lee et al. [93] | ✓ | ✓ | ✓ | ✓ | ✓ | ✓ | ✓ | - | | Excellent |
| Khazaei et al. [94] | ✓ | ✓ | ✓ | ✓ | ✓ | - | - | - | | Good |
| Dehghani et al. [95] | ✓ | ✓ | ✓ | ✓ | - | - | - | ✓ | | Good |
| Yeh et al. [96] | ✓ | - | ✓ | - | ✓ | - | - | - | | Fair |
| Kang et al. [97] | ✓ | ✓ | ✓ | ✓ | ✓ | ✓ | - | - | | Excellent |
| Tutu et al. [98] | ✓ | - | ✓ | ✓ | - | - | - | - | | Fair |
| Cardoso et al. [99] | ✓ | ✓ | ✓ | - | - | - | - | - | | Fair |
| De Sousa et al. [100] | ✓ | ✓ | ✓ | - | - | ✓ | - | - | | Good |
| Li et al. [101] | ✓ | ✓ | ✓ | ✓ | ✓ | - | - | ✓ | | Excellent |
| Wu et al. [102] | ✓ | ✓ | ✓ | - | ✓ | - | - | - | | Good |
| Martins et al. [103] | - | ✓ | ✓ | - | - | - | - | - | | Fair |

(*Continued*)

**Table 8.** (Continued)

| Author [ref.] | Reliability | | Content & face | Validity | | | | |  | Ratings |
| | Internal Consistency | Test-retest (ICC) | | Structural | | Criterion | Hypothesis testing | | |
| | | | | EFA | CFA | Predictive & Concurrent | Convergent | Discrimination & Known groups comparison | |
| Echeverri et al. [104] | ✓ | - | ✓ | ✓ | ✓ | - | - | ✓ | Good |
| Huang et al. [105] | ✓ | - | | - | - | - | - | - | Poor |
| Rajabi et al. [106] | ✓ | - | ✓ | ✓ | - | - | - | - | Fair |
| Wei et al. [107] | ✓ | - | ✓ | - | ✓ | - | - | - | Fair |
| Chen et al. [108] | ✓ | ✓ | ✓ | ✓ | ✓ | - | - | - | Good |
| Savci et al. [109] | ✓ | - | ✓ | ✓ | ✓ | ✓ | - | - | Good |
| Hiltrop et al. [110] | ✓ | - | - | ✓ | ✓ | - | ✓ | - | Good |
| **Content specific health literacy instruments** | | | | | | | | | |
| Cormier et al. [111]* | ✓ | - | ✓ | ✓ | - | - | - | - | Fair |
| Sabbahi et al. [112] | ✓ | ✓ | ✓ | - | - | ✓ | ✓ | ✓ | Excellent |
| Kumar et al. [113] | ✓ | - | ✓ | - | - | - | - | ✓ | Fair |
| Macek et al. [114] | ✓ | - | ✓ | - | - | ✓ | - | - | Fair |
| Devi et al. [115] | ✓ | ✓ | - | - | - | ✓ | ✓ | - | Good |
| Mojoyinola [116] | ✓ | - | - | - | - | - | - | - | Poor |
| Loureiro et al. [117] | ✓ | - | - | ✓ | - | - | - | - | Fair |
| Wong et al. [118] | ✓ | ✓ | ✓ | - | - | ✓ | ✓ | - | Good |
| Dahlke et al. [119] | - | - | ✓ | - | - | ✓ | ✓ | - | Fair |
| Jones et al. [120] | ✓ | ✓ | ✓ | ✓ | - | ✓ | ✓ | ✓ | Excellent |
| Naghibi Sistani et al. [121] | ✓ | ✓ | ✓ | - | - | - | - | ✓ | Good |
| Paez et al. [122] | ✓ | - | - | ✓ | ✓ | - | ✓ | - | Good |
| Shreffler-Grant et al. [123] | ✓ | - | ✓ | ✓ | - | - | ✓ | - | Good |
| Villanueva Vilchis et al. [124] | ✓ | ✓ | ✓ | - | - | - | ✓ | - | Good |
| O'Connor et al. [125] | ✓ | ✓ | ✓ | ✓ | - | ✓ | - | ✓ | Excellent |
| Altin et al. [126] | ✓ | - | - | ✓ | ✓ | ✓ | - | - | Good |
| Curtis et al. [127] | ✓ | - | - | - | ✓ | ✓ | ✓ | - | Good |
| Guttersrud et al. [128] | ✓ | - | - | - | - | - | - | - | Poor |
| Stein et al. [129] | ✓ | ✓ | ✓ | - | - | ✓ | - | - | Good |
| Intarakamhanga et al. [130] | ✓ | - | ✓ | ✓ | ✓ | - | - | - | Good |
| Kapoor et al. [131] | ✓ | ✓ | ✓ | - | - | ✓ | ✓ | - | Good |
| Jung et al. [132] | ✓ | - | ✓ | ✓ | ✓ | - | ✓ | ✓ | Excellent |
| Campos et al. [133] | ✓ | ✓ | ✓ | ✓ | - | - | - | - | Good |
| Squires et al. [134] | ✓ | - | ✓ | ✓ | - | - | - | ✓ | Fair |
| Bjornsen et al. [135] | ✓ | ✓ | ✓ | ✓ | ✓ | - | - | ✓ | Excellent |
| Moll et al. [136] | ✓ | - | - | ✓ | - | - | ✓ | ✓ | Good |
| Intarakamhang et al. [137] | ✓ | - | - | ✓ | ✓ | - | - | - | Fair |
| Matsumoto et al. [138] | ✓ | - | ✓ | - | ✓ | - | - | - | Fair |
| Tsaia et al. [139] | ✓ | - | - | ✓ | ✓ | ✓ | ✓ | - | Good |
| Lichtveld et al. [140] | ✓ | - | ✓ | ✓ | ✓ | - | - | - | Good |
| Areerak et al. [141] | ✓ | ✓ | ✓ | ✓ | ✓ | - | - | ✓ | Excellent |

(*Continued*)

**Table 8.** (Continued)

| Author [ref.] | Reliability | | Content & face | Validity | | | | | | Ratings |
| | | | | Construct | | | | | | |
| | | | | Structural | | Criterion | Hypothesis testing | | | |
| | Internal Consistency | Test- retest (ICC) | | EFA | CFA | Predictive & Concurrent | Convergent | Discrimination & Known groups comparison | | |
| Zhang et al. [142] | ✓ | ✓ | ✓ | - | ✓ | - | - | - | | Good |
| Irvin et al. [143] | ✓ | - | ✓ | ✓ | - | ✓ | - | ✓ | | Good |
| Wei et al. [144] | ✓ | - | ✓ | ✓ | - | - | - | ✓ | | Good |
| Ayre et al. [145] | ✓ | - | ✓ | - | ✓ | ✓ | - | - | | Good |
| Intarakamhang et al. [146] | ✓ | - | ✓ | - | ✓ | - | - | - | | Fair |
| Suthakorn et al. [147] | ✓ | - | ✓ | ✓ | ✓ | - | - | - | | Good |
| Lin et al. [148] | ✓ | - | ✓ | ✓ | - | - | ✓ | ✓ | | Good |
| Taheri et al. [149] | ✓ | ✓ | ✓ | ✓ | - | - | - | - | | Good |
| Tabacchi et al. [150] | ✓ | - | ✓ | - | ✓ | - | - | ✓ | | Good |
| Zenasa et al. [151] | ✓ | - | ✓ | ✓ | ✓ | - | - | - | | Good |
| Taoufik et al. [152] | ✓ | ✓ | ✓ | - | - | - | ✓ | - | | Good |
| Chao et al. [153] | ✓ | - | ✓ | ✓ | ✓ | - | ✓ | ✓ | | Excellent |
| Sun et al. [154] | ✓ | ✓ | ✓ | ✓ | - | ✓ | - | ✓ | | Excellent |
| Poureslami et al. [155] | - | - | ✓ | - | - | - | - | - | | Poor |
| Mahmoudian et al. [156] | ✓ | - | ✓ | - | - | - | - | - | | Fair |
| Simkiss et al. [157] | ✓ | ✓ | ✓ | ✓ | ✓ | - | - | - | | Good |
| Charophasrat et al. [158] | ✓ | - | ✓ | - | - | ✓ | | ✓ | | Good |
| Karimi et al. [159] | ✓ | ✓ | ✓ | - | - | - | - | - | | Fair |
| Ma et al. [160] | ✓ | ✓ | ✓ | - | ✓ | - | - | - | | Good |
| Suto et al. [161] | ✓ | - | ✓ | ✓ | - | ✓ | - | - | | Good |
| Kodama et al. [162] | ✓ | ✓ | ✓ | ✓ | ✓ | ✓ | - | - | | Excellent |
| Aller et al. [163] | ✓ | - | ✓ | ✓ | - | - | ✓ | - | | Good |
| Robbins et al. [164] | ✓ | - | - | - | - | - | - | - | | Poor |
| Rabin et al. [165] | ✓ | - | ✓ | - | - | - | - | ✓ | | Fair |
| Moein et al. [166] | ✓ | ✓ | ✓ | ✓ | ✓ | - | - | - | | Good |
| **Population-specific health literacy instruments** | | | | | | | | | | |
| Lee TW et al. [167] | ✓ | - | ✓ | ✓ | ✓ | - | - | - | | Good |
| Pan et al. [168] | ✓ | - | - | - | - | ✓ | - | ✓ | | Fair |
| Tsai et al. [169] | ✓ | - | ✓ | ✓ | ✓ | ✓ | ✓ | - | | Excellent |
| Weidmer et al. [170] | ✓ | - | - | - | ✓ | - | - | - | | Fair |
| Massey et al. [171] | ✓ | - | ✓ | ✓ | - | - | - | - | | Fair |
| Wang et al. [172] | ✓ | - | ✓ | ✓ | ✓ | - | - | - | | Good |
| Harper et al. [173] | - | - | ✓ | - | ✓ | - | - | - | | Fair |
| Yuen et al. [174] | - | - | ✓ | - | - | - | - | - | | Poor |
| Manganello et al. [175] | ✓ | - | ✓ | ✓ | - | ✓ | - | - | | Good |
| Shen et al. [176] | ✓ | - | - | - | ✓ | - | - | ✓ | | Fair |
| Abel et al. [177] | ✓ | - | - | ✓ | ✓ | - | - | ✓ | | Good |
| Ghanbari et al. [178] | ✓ | ✓ | ✓ | ✓ | - | - | - | - | | Good |
| Paakkari et al. [179] | ✓ | ✓ | ✓ | - | ✓ | - | - | - | | Good |
| Yang et al. [180] | ✓ | - | - | - | ✓ | ✓ | - | ✓ | | Good |
| Ernstmann et al. [181] | ✓ | - | - | ✓ | ✓ | - | - | - | | Fair |

(*Continued*)

**Table 8.** (Continued)

| Author [ref.] | Reliability | | Content & face | Validity | | | | | | Ratings |
| | | | | Content & face | Construct | | | | | |
| | | | | | Structural | | Criterion | Hypothesis testing | | |
| | Internal Consistency | Test- retest (ICC) | | EFA | CFA | Predictive & Concurrent | Convergent | Discrimination & Known groups comparison | |
| Chang et al. [182] | ✓ | - | - | ✓ | ✓ | - | ✓ | ✓ | Good |
| Eliason et al. [183] | ✓ | ✓ | ✓ | ✓ | - | - | - | - | Good |
| Hashimoto et al. [184] | ✓ | - | ✓ | ✓ | ✓ | ✓ | - | - | Good |
| Bradley-Klug et al. [185] | ✓ | - | - | ✓ | - | - | - | ✓ | Fair |
| Guo et al. [186] | ✓ | - | ✓ | - | ✓ | - | ✓ | - | Good |
| Azizi et al. [187] | ✓ | ✓ | ✓ | ✓ | - | - | - | - | Good |
| Domanska et al. [188] | ✓ | - | ✓ | - | ✓ | ✓ | ✓ | - | Good |
| **Electronic health literacy instruments** | | | | | | | | | |
| Norman et al. [189] | ✓ | ✓ | ✓ | ✓ | - | - | - | - | Good |
| Hahn et al. [190] | ✓ | - | ✓ | - | - | - | - | ✓ | Fair |
| Ownby et al. [191] | ✓ | - | ✓ | ✓ | - | ✓ | - | ✓ | Good |
| Seçkin et al. [192] | ✓ | - | - | ✓ | ✓ | - | - | - | Fair |
| Van der Vaart et al. [193] | ✓ | ✓ | ✓ | ✓ | - | - | - | - | Good |
| Kayser et al. [194] | ✓ | - | - | ✓ | ✓ | - | - | - | Fair |
| Paige et al. [195] | ✓ | - | - | - | ✓ | - | - | - | Fair |
| Woudstra et al. [196] | ✓ | - | - | - | ✓ | ✓ | ✓ | - | Good |
| Castellvi et al. [197] | ✓ | ✓ | - | - | - | - | ✓ | ✓ | Good |
| Liu et al. [198] | ✓ | ✓ | ✓ | ✓ | ✓ | ✓ | ✓ | - | Excellent |
| Duong et al. [199] | ✓ | - | ✓ | ✓ | - | - | ✓ | - | Good |

**Table 9. A list of instruments that well developed, reported proper psychometric properties, and instruments frequently translated or validated in other countries (1993–2021).**

| | Instruments |
| --- | --- |
| **Well-developed instruments** | |
| | Health Literacy Questionnaire (HLQ) (validity-driven) [55] |
| | Health Literacy Survey Questionnaire (HLS-EU-Q-47) (conceptual-based, multi-faceted attributes) [56] |
| **Instruments with excellent reported psychometric properties** | |
| | Chinese Health Literacy Scale for Low Salt Consumption-Hong Kong population (CHLSalt-HK) [64] |
| | Comprehensive Diabetes Health Literacy Scale (DHLS) [93] |
| | Korean Health Literacy Scale for Diabetes Mellitus (KHLS-DM) [97] |
| | Chinese Health Literacy Scale for Tuberculosis (CHLS-TB) [101] |
| | Oral Health Literacy Instrument (OHLI) [112] |
| | Health Literacy in Dentistry scale (HeLD-29) [120] |
| | Mental Health Literacy Scale (MHLS) [125] |
| | Multicomponent mental health literacy measure [132] |
| | Mental Health-Promoting Knowledge (MHPK-10) [135] |

(Continued)

**Table 9.** (Continued)

| | Instruments |
|---|---|
| | Neck pain-specific Health Behavior in Office Workers (NHBOW) [141] |
| | Mental Health Literacy Scale for Healthcare Students (MHLS-HS) [153] |
| | The Comprehensive Oral Health Literacy (COHL) [154] |
| | Mental Health Literacy Scale for Depression Affecting the Help-Seeking Process [162] |
| | Mandarin Health Literacy Scale (MHLS) [169] |
| | eHealth Literacy Scale (eHLS-Web 3.0) [198] |
| **Frequently translated or validated (more than ten)** | |
| | Rapid Estimate of Adult Literacy in Medicine (REALM) [14] |
| | Short form of the Test of Functional Health Literacy in Adults (S-TOFHLA) [41] |
| | Newest Vital Sign (NVS) [16] |
| | Functional, Communicative, and Critical Health Literacy (FCCHL) [45] |
| | Health Literacy Questionnaire (HLQ) [55] |
| | European Health Literacy Survey Questionnaire (HLS-EU-Q-47) [56] |
| | Short versions of the European Health Literacy Survey Questionnaire (HLS-EU-Q16, Q6) [61] |
| | e-Health Literacy Scale (e-HEALS) [189] |

Organization should take responsibility and provide a clear guideline for measuring health literacy as appropriate.

## Supporting information

**S1 Checklist. PRISMA 2020 checklist.**
(DOCX)

## Acknowledgments

The authors are grateful to all staff in Iranian Institute for Health Sciences Research, ACECR, Tehran, Iran for help and support.

## Author Contributions

**Conceptualization:** Mahmoud Tavousi, Jila Sadighi, Ali Montazeri.

**Data curation:** Mahmoud Tavousi, Samira Mohammadi, Fatemeh Zarei.

**Formal analysis:** Mahmoud Tavousi, Ali Montazeri.

**Investigation:** Samira Mohammadi, Fatemeh Zarei, Ramin Mozafari Kermani, Rahele Rostami.

**Methodology:** Mahmoud Tavousi, Samira Mohammadi, Jila Sadighi, Ali Montazeri.

**Supervision:** Ali Montazeri.

**Writing – original draft:** Mahmoud Tavousi, Samira Mohammadi, Fatemeh Zarei.

**Writing – review & editing:** Ali Montazeri.

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
