## [Decision Letter · Decision Letter 0]

2 May 2022

PONE-D-22-04494Measuring health literacy: A systematic review and bibliometric analysis of instruments from 1993 to 2021PLOS ONE

Dear Dr. Ali Montazeri,

Thank you for submitting your manuscript to PLOS ONE. After careful consideration, we feel that it has merit but does not fully meet PLOS ONE’s publication criteria as it currently stands. Therefore, we invite you to submit a revised version of the manuscript that addresses the points raised during the review process.

We look forward to receiving your revised manuscript.

Kind regards,

Carlos Miguel Rios-González, Ph.D

Academic Editor

PLOS ONE

Journal Requirements:

Reviewers' comments:

Reviewer's Responses to Questions

**Comments to the Author**

1. Is the manuscript technically sound, and do the data support the conclusions?

Reviewer #1: Yes

2. Has the statistical analysis been performed appropriately and rigorously? 

Reviewer #1: Yes

3. Have the authors made all data underlying the findings in their manuscript fully available?

Reviewer #1: Yes

4. Is the manuscript presented in an intelligible fashion and written in standard English?

Reviewer #1: No

5. Review Comments to the Author

Reviewer #1: Professor Montazeri’s and colleague’s study appears to do something quite unusual: reviewing the literature on measurement instrument construction for the or concept of health literacy. His own sentence differs slightly: “This study aimed to review all existing instruments to summarize the current knowledge on the topic .” This reads to me as if either the development of measurement instruments was the subject of the review, or the instruments themselves or rather their interrelationship with outcomes, the precise nature of which would remain in the dark,

The results and conclusions formulate rather unspectacular assertions, such as there are enough suitable measures of health literacy, there is a strong tradition of disease-specific, group-specific and technology-specific instruments, A minor problem is that of choosing the right measure. The authors forward the idea that the WHO might set some standards for that.

The question arises: Do we need a costly systematic review for results and conclusions like those mentioned. And if we go back to the original ideas behind systematic reviewing, One of the purposes of systematic reviewing is to come to a decision if there is evidence, say, for the efficacy of treatment A as well as for treatment B. If review studies of quality exist on the issue, they will tell you which is better, A or B. Nothing of this urgency do we find in the study to be reviewed. And moreover, the authors do not review all of the evidence, deliberately. They include only publications about the development of a measure, or its transfer to another country,

Study presentation, explanation, structure and language are fine, though.

6. PLOS authors have the option to publish the peer review history of their article (what does this mean?). If published, this will include your full peer review and any attached files.

Reviewer #1: No

---

## [Author Response · Author response to Decision Letter 0]

14 May 2022

Thank you. Done.

Data availability was revised and now reads as follows:

Data Availability 

The authors confirm that all data underlying the findings are fully available without restriction. All relevant data are within the paper.

Review Comments to the Author

Reviewer #1 

Professor Montazeri’s and colleague’s study appears to do something quite unusual: reviewing the literature on measurement instrument construction for the or concept of health literacy. His own sentence differs slightly: “This study aimed to review all existing instruments to summarize the current knowledge on the topic.” This reads to me as if either the development of measurement instruments was the subject of the review, or the instruments themselves or rather their interrelationship with outcomes, the precise nature of which would remain in the dark.

Thank you for your comment. The sentence was revised.

This study aimed to review all existing instruments to summarize the current knowledge on the development of existing measurement instruments and their possible translation and validation in other languages different from the original languages.

The results and conclusions formulate rather unspectacular assertions, such as there are enough suitable measures of health literacy, there is a strong tradition of disease-specific, group-specific and technology-specific instruments, a minor problem is that of choosing the right measure. The authors forward the idea that the WHO might set some standards for that.

The question arises: Do we need a costly systematic review for results and conclusions like those mentioned. And if we go back to the original ideas behind systematic reviewing, one of the purposes of systematic reviewing is to come to a decision if there is evidence, say, for the efficacy of treatment A as well as for treatment B. If review studies of quality exist on the issue, they will tell you which is better, A or B. Nothing of this urgency do we find in the study to be reviewed. And moreover, the authors do not review all of the evidence, deliberately. They include only publications about the development of a measure, or its transfer to another country, Study presentation, explanation, structure and language are fine, though.

Thank you for your thoughtful comment. It was useful, and it has led us to think more carefully in summarizing the evidence. Indeed, we now fully revised the following section and added a table for more clarity. Also, we rewrote the previous synthesis of findings and moved it with some corrections to the end of the discussion. We also revised the conclusion to reflect the critics by the respected reviewer:

A: synthesis of findings

Numerous instruments have been developed during the past thirty years for measuring health literacy. This review could provide information on 162 instruments. Of these, there were two well-developed instruments: 

1. HLQ, which avoided the use of prevailing theories until the later development process, and great care was taken to fully understand the experiences and lives of people, professionals, and healthcare providers [55].

2. HLS-EU-Q47, which used conceptual-based, multi-faceted attributes [56]. 

However, they reported limited psychometric properties. Of the remaining instruments, 15 instruments reported proper psychometric properties needed. In addition, there were a number of instruments that were translated and validated to other languages more frequently. A list of instruments is presented in Table 9.

B: Discussion 

Finally, one should note that the most important question is, do we need so many instruments for measuring health literacy? Although one could not prevent investigators from developing new instruments, it is evident that such haphazard development of instruments is not helpful. It seems that we need a core global general health literacy instrument for use around the globe. Then perhaps it is possible to add a few contents/disease-specific, population-specific, or e-health literacy items to the general instruments according to their use. The experience of the European Organization for Research and Treatment of Cancer-EORTC (the Quality of Life Study Group) might be useful to be adapted (https://qol.eortc.org/quality-of-life-group/).

C: Conclusion

This review highlighted that there were more than enough instruments for measuring health literacy. In addition, we found that a number of instruments did not report psychometric properties sufficiently. However, evidence suggest that well developed instruments and those reported adequate measures of validation could be helpful if appropriately selected based on objectives of a given study. Perhaps an authorized institution such as World Health Organization should take responsibility and provide a clear guideline for measuring health literacy as appropriate.

---

## [Editor Report · Decision Letter 1]

5 Jul 2022

Measuring health literacy: A systematic review and bibliometric analysis of instruments from 1993 to 2021

PONE-D-22-04494R1

Dear Dr. Ali Montazeri,

We’re pleased to inform you that your manuscript has been judged scientifically suitable for publication and will be formally accepted for publication once it meets all outstanding technical requirements.

Kind regards,

Carlos Miguel Rios-González, Ph.D

Academic Editor

PLOS ONE

---

## [Editor Report · Acceptance letter]

7 Jul 2022

PONE-D-22-04494R1 

Measuring health literacy: A systematic review and bibliometric analysis of instruments from 1993 to 2021 

Dear Dr. Montazeri:

I'm pleased to inform you that your manuscript has been deemed suitable for publication in PLOS ONE. Congratulations! Your manuscript is now with our production department. 

Kind regards, 

on behalf of

Dr. Carlos Miguel Rios-González 

Academic Editor

PLOS ONE